# The Efficiency Gap in Byte Modeling

**Celine Lee** [2 3]   **Jing Nathan Yan** [1]   **Chen Liang** [1]   **Jiaxin Shi** [2]   **Yin Zhang** [1]   **Jeremiah Liu** [2]   **Pengcheng Yin** [2]
**Fernando Pereira** [2]   **Ed Chi** [1]   **Derek Cheng** [1]   **Alexander M. Rush** [3]   **Ruoxi Wang** [1]

## Abstract

Modern language models have historically relied on two dominant design choices: subword tokenization and autoregressive (AR) ordering. These design decisions bake in priors that dictate a model's learning. Recently, two alternative paradigms have challenged this: byte-level modeling, which bypasses static statistically-derived token vocabularies, and masked diffusion modeling (MDM), which conducts parallel, non-sequential generation. Their intersection represents a fully end-to-end modality-agnostic generative prototype; however, removing these structural priors incurs a significant computational cost. In this work, we investigate this cost through a compute-matched scaling study. Our results reveal that the performance penalty of byte modeling is not uniform; across scale, the scaling overhead of byte modeling is worse for MDM than for AR. We hypothesize that this disparity stems from context fragility: while AR's stable causal history allows models to naturally rediscover subword patterns, the MDM objective destroys the local contiguity required to efficiently resolve semantics from raw bytes. Our findings from controlled permutation experiments suggest that future modality-agnostic designs must incorporate alternative structural biases to maintain viable scaling trajectories in the byte regime.

## 1. Introduction

While rapid scaling has unlocked remarkable capabilities in large language models (LLMs) (Brown et al., 2020; Hoffmann et al., 2022; Kaplan et al., 2020; Wei et al., 2022; OpenAI et al., 2024), the standard recipe remains anchored

to subword tokenization and autoregressive (AR) ordering, which impose specific structural limitations. Subword tokenization (e.g., BPE (Sennrich et al., 2016), Wordpiece (Devlin et al., 2019)) acts as a fixed compression layer that establishes a domain-specific prior that can hinder generalization to out-of-distribution modalities (Xue et al., 2022). Simultaneously, the AR objective enforces a unidirectional bias, which may weaken the look-ahead planning required for non-sequential reasoning (Nie et al., 2025).

Two alternative paradigms present opportunities to bypass these constraints: **byte-level modeling** for universality (YU et al., 2023; Hwang et al., 2025; Wang et al., 2024; Pagnoni et al., 2024), and **masked diffusion models (MDMs)** for order-agnostic inference (Sahoo et al., 2024; Shi et al., 2024). While their intersection may represent a generative ideal for fine-grained any-order modeling, their computational interaction remains under-explored. This work investigates the inherent computational cost incurred when these linguistic and structural priors are removed.

In this work, we quantify the scaling overhead of byte modeling through a compute-matched study and find that it is objective-dependent. To isolate the interaction between modeling objectives and data representation, we utilize a standard Transformers (Vaswani et al., 2017) backbone as a controlled setup to bypass confounding architectural variables. Our results reveal that byte modeling introduces a scaling overhead distinct from the Transformers' quadratic attention cost: the model must expend additional compute to process subword structures that BPE provides for free.

While AR byte models quickly approach performance parity with their BPE counterparts, byte-level MDMs exhibit a longer-sustained performance offset. Our isoFLOPs extrapolation suggests that while AR reaches parity within FLOPs budgets $F \approx 10^{22}$, MDMs require budgets as high as $F \approx 4 \times 10^{26}$ to close this gap. Furthermore, the relative rate of improvement for byte models is less for MDM than for AR. This sustained overhead indicates that parallel-generation objectives demand a significantly higher computational investment to resolve sequences without the guidance of pre-composed semantic units.

To investigate these empirical results, we conduct a mechanistic investigation into the roots of this disparity. We

---
[1]Google DeepMind [2]Work done while at Google DeepMind [3]Department of Computer Science, Cornell University. Correspondence to: Celine Lee <cl923@cornell.edu>.

*Proceedings of the 43rd International Conference on Machine Learning*, Seoul, South Korea. PMLR 306, 2026. Copyright 2026 by the author(s).

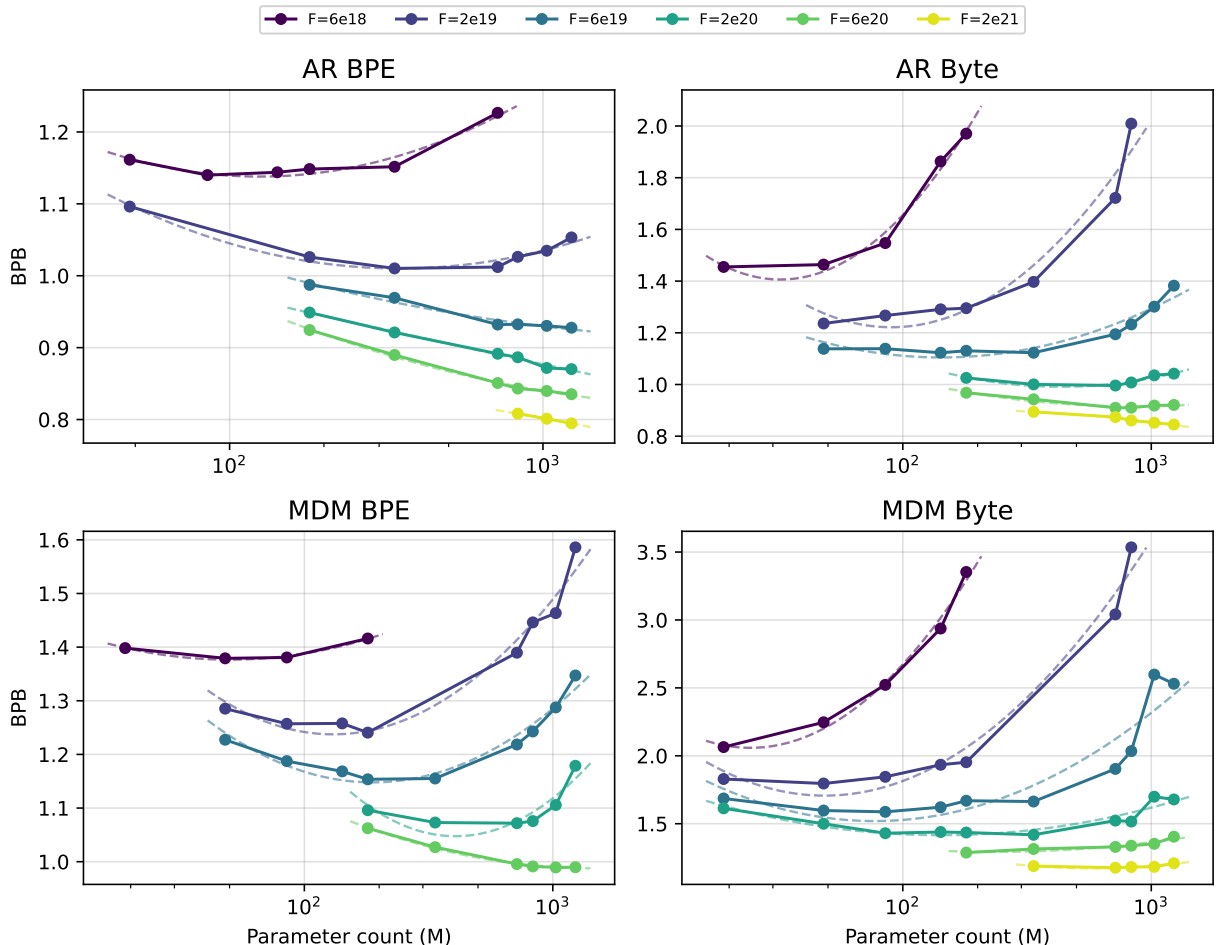

*Figure 1.* IsoFLOPs curves are shown for autoregressive (AR, top) and masked diffusion (MDM, bottom) objectives with BPE (left) and byte (right) tokenizers. We evaluate models from 48M to 1.2B non-embedding parameters using Bits-per-Byte (BPB) as a unified metric. Each curve represents a fixed training compute budget ranging from $F = 6 \times 10^{18}$ to $F = 2 \times 10^{21}$ FLOPs, with dotted parabolas approximating the efficiency frontier for each budget. While the transition from BPE to raw bytes shifts the BPB higher for both objectives, AR byte models converge toward BPE counterparts at scale, whereas MDM byte models exhibit a persistent performance offset.

identify emergent segmentation in causal models, where AR byte predictive entropy aligns with BPE boundaries, indicating that the model's efficiency is driven by its ability to resolve inter-byte transitions within a wordpiece once the first few bytes are established. This advantage is lost in the stochastic masking of MDM, which deprives the model of a guaranteed context helpful to form rich prediction latents. We decompose this context fragility through controlled permutation experiments that isolate the local and sequential context destruction of non-causal generation. These experiments confirm that byte-level modeling is uniquely sensitive to sequential integrity: this is a fundamental structural mismatch in byte-level text diffusion. Future modality-agnostic designs must incorporate alternative structural priors to maintain viable scaling trajectories.

## 2. Related Work

**Byte-Level Language Models.** The goal of universal, tokenizer-free language modeling has driven recent explorations into bypassing BPE in favor of raw UTF-8 bytes. To manage the extreme sequence lengths inherent to byte streams, recent architectures employ hierarchical patching (Pagnoni et al., 2024; YU et al., 2023) or linear-time backbones like state space models (Wang et al., 2024). Hybrid approaches such as H-Nets (Hwang et al., 2025) leverage learned byte grouping to bridge the gap between low-level signal and high-level expressivity. However, these works operate within the autoregressive paradigm, benchmarking efficiency against standard BPE Transformers in left-to-right generation. In this work, we demonstrate that the viability of bypassing pre-model subword tokenization is not universal but is in fact contingent on the modeling objec-

tive, a dependency that has remained largely uninvestigated in prior byte-level literature.

**Diffusion Language Models.** Diffusion models, originally dominant in computer vision, have been successfully adapted to discrete text generation. D3PM (Austin et al., 2023) introduced the discrete diffusion framework with transition matrices. MDLM (Sahoo et al., 2024) and MD4 (Shi et al., 2024) simplified this into a masked prediction objective that is competitive with AR baselines. Other approaches like block diffusion (Arriola et al., 2025) explore modeling modifications that interpolate between autoregressive and diffusion modeling to enable flexible-length generation. Eso-LMs (Sahoo et al., 2025) further present a hybrid approach in which a masked diffusion language model diffuses tokens in parallel, followed by a sequential stage that unmasks the remaining tokens sequentially. Architectural innovations such as DiffuApriel (Singh et al., 2025) have introduced bidirectional Mamba backbones to improve the throughput of these denoisers by making them linear-time rather than quadratic. Despite these advancements, these models predominantly rely on subword tokenization. While small-scale character-level experiments exist (e.g. on text8), they lack the systematic scaling analysis required to understand the representational challenges of non-causal objectives on raw bytes.

**Scaling Laws for Compute and Vocabulary.** The characterization of model performance as a function of compute budget was formalized by Hoffman et al. (Hoffmann et al., 2022), establishing the "Chinchilla" scaling laws for optimal parameter and data allocation. Recent investigations into masked diffusion scaling suggest that while MDMs exhibit loss decrease rates comparable to AR models (Nie et al., 2025), they maintain a persistent computational gap of approximately $16\times$ that of AR models. Follow-up work (von Rütte et al., 2026) finds that MDMs may require a higher parameters-to-data ratio at large scales to achieve parity with AR baselines.

Relatedly, scaling laws for vocabulary (Tao et al., 2024) indicate that vocabulary size plays an important role in optimizing model performance: larger semantic units (i.e. larger-vocabulary tokenizers) help larger models. Smaller vocabularies have been noted to increase the complexity of the denoising task in masked diffusion, likely due to the resulting increase in long-range dependencies (Sahoo et al., 2024). Our work explicitly studies the impact of data representation and modeling objective on performance with scale, finding that token representation interacts with modeling objective beyond simple context length: the order-agnostic nature of MDM makes it uniquely dependent on the pre-composed semantic units of subword tokens.

## 3. Background

### 3.1. Language Modeling Objectives

Language modeling is the task of learning the probability distribution of sequences $x = (x_1, x_2, ...x_L)$. The choice of factorization and generation order distinguishes autoregressive (AR) models from masked diffusion models (MDMs).

**Autoregressive Modeling.** AR models factorize the joint probability as a product of conditional probabilities using a fixed left-to-right order:

$$p_\theta(x) = \prod_{i=1}^{L} p_\theta(x_i|x_{<i}),$$

parameterized by model $\theta$. This formulation enforces a strict sequential dependency, where the model attends only to past tokens when predicting the next token.

**Masked Diffusion Modeling.** Unlike AR models, MDMs are inherently bidirectional and order-agnostic. We follow the continuous-time discrete diffusion framework (Austin et al., 2023; Shi et al., 2024; Sahoo et al., 2024), where a forward process independently replaces tokens in $x_0$ with [MASK] based on a retention schedule $\alpha_t$. The model learns to reverse this by minimizing the weighted objective:

$$\mathcal{L}_{\text{MDM}} = \mathbb{E}_{t,x_t} \left[ \frac{\alpha'_t}{1-\alpha_t} \sum_{i:x_t^{(i)}=\text{[MASK]}} \log p_\theta(x^{(i)}|x_t, t) \right],$$

which prioritizes mostly-unmasked states. Inference entails iterative unmasking. Crucially, whereas AR models rely on a stable causal history, MDMs must resolve semantics from a combinatorial set of fragmented contexts. This distinction is central to our investigation into how byte-level representations, which lack pre-composed BPE units, affect scaling across paradigms. More details can be found in Appendix A.

### 3.2. BPE Compression

Byte-Pair Encoding (BPE) (Gage, 1994; Sennrich et al., 2016) is the dominant tokenization strategy in modern LLMs. BPE iteratively merges the most frequent adjacent pairs of bytes into new tokens, building a vocabulary $V$ of subword units. This process compresses the raw data: a sequence of bytes is represented by fewer BPE tokens.

This compression has two primary effects: (1) **computational efficiency**, as the attention mechanism scales quadratically with sequence length $L$ and (2) **semantic density**.

BPE tokens often correspond to morphemes, whole words, or common sub-word structures, meaning each individual token carries higher information content. In contrast, byte-level modeling involves no compression, resulting in longer sequences where individual units (bytes) carry minimal semantic value in isolation.

## 4. Experimental Setup

To isolate the interaction between data representation (byte vs. BPE) and modeling objective (AR vs. MDM), we employ a compute-matched evaluation protocol (Hoffmann et al., 2022) that ensures comparisons are grounded in total floating-point operations (FLOPs) expended.

### 4.1. Data Preparation

**Pre-training Data.** All models are pre-trained on the Slimpajama-627B dataset (Soboleva et al., 2023).

**Tokenization Strategies.** We compare two distinct input representations: **(1) byte-level:** map raw UTF-8 bytes directly to a vocabulary of size $V = 256$; **(2) BPE-level:** use the standard Llama 2 BPE tokenizer (Touvron et al., 2023), which has a vocabulary size of $V = 32,000$.

**Sequence Length Normalization.** To ensure all model types process the same volume of raw data per optimization step, we normalize the information content of the context window. Byte models are trained with a context window of $L_{byte} = 8192$ raw bytes, and BPE models with $L_{BPE} = 1792$ tokens.

### 4.2. Model Training

**Model Architecture.** We evaluate models across a parameter sweep from 48M to 1.23B non-embedding parameters, using a standard Transformer architecture.

All models utilize pre-normalization, SwiGLU activation functions, and RoPE embedding with full global attention. Autoregressive models use a causal masking mechanism. Masked diffusion models use bidirectional attention to allow global context reasoning during the denoising process.

**MDM training schedule.** For MDM training, we train according to a linear masking schedule:

$$\alpha_t = 1 - t; \frac{\alpha_t'}{1 - \alpha_t} = -\frac{1}{t},$$

and evaluate with a cosine masking schedule:

$$\alpha_t = 1 - cos\left(\frac{\pi}{2}(1 - t)\right); \frac{\alpha_t'}{1 - \alpha_t} = -\frac{\pi}{2}tan\left(\frac{\pi}{2}(1 - t)\right),$$

in accordance with best practices from Shi et al. (2024).

**Optimization.** Models are trained using the AdamW optimizer (Loshchilov & Hutter, 2019) ($\beta_1 = 0.9$, $\beta_2 = 0.95$) with a global batch size of $B = 1,152$. Learning rate is swept logarithmically from 1e-4 to 3e-3 with a $1\%$ linear warm-up followed by cosine decay to minimum learning rate 2e-4. Weight decay is set to 0.1; gradient clipping is swept between 0.25 and 1.0.

### 4.3. Evaluation Metric: Bits-Per-byte (BPB)

To compare across divergent vocabulary sizes on a unified scale, we report **Bits-Per-byte (BPB)**. This metric normalizes the total log-likelihood by the raw size of the dataset in bytes, effectively decoupling predictive performance from the discretization strategy. Formally:

$$BPB = \frac{NLL(D; \theta)/|D|_{\text{bytes}}}{ln(2)},$$

where $NLL(D; \theta) = -\sum_{x \in D} \log p_\theta(x)$ represents the total negative log-likelihood of the *tokenized* dataset and $|D|_{\text{bytes}}$ is byte count of the raw data.

### 4.4. Evaluation Protocols

**Compute-Matched Comparison.** Due to the $O(L^2)$ complexity of the attention mechanism, byte-level models consume significantly more FLOPs per training step than BPE models of equivalent parameter count. To remain matched on total training FLOPs, we adjust the data budget of byte models to accommodate this compute imbalance. We train models across a sweep of fixed compute budgets ($F \in \{6e18, 2e19, 6e19, 2e20, 6e20, 2e21\}$) and compute the data budget for each model size accordingly. Detailed FLOPs computations are provided in Appendix B.

**Downstream Task Evaluation.** To verify that BPB disparities translate to semantic capabilities, we evaluate specific model pairs on downstream reasoning benchmarks: ARC-Easy (Clark et al., 2018), BoolQ (Clark et al., 2019), HellaSwag (Zellers et al., 2019), OBQA (Mihaylov et al., 2018), PIQA (Bisk et al., 2020), RACE (Lai et al., 2017), and SIQA (Sap et al., 2019). We define two matching protocols to isolate the effects of computational investment and model capacity:

**Compute match:** We select models that have consumed equivalent training FLOPs ($F \approx 2 \times 10^{20}$). Due to the quadratic cost of the $4\times$ longer byte sequences, this protocol compares a lower-capacity byte model against a higher-capacity BPE baseline (e.g. 180M byte vs. 717M BPE).

**Capacity match:** We isolate representational differences by comparing models with identical non-embedding parameter counts (e.g. 717M byte vs. 717M BPE) trained on equal data volumes. This highlights the performance

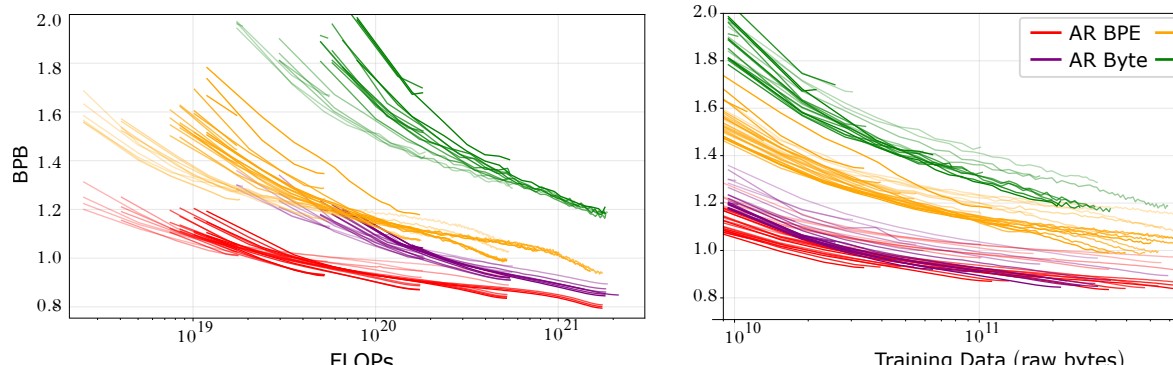

*Figure 2.* **Training curves across objectives. FLOPs (Left):** BPB plotted against total training compute show that while AR byte models (purple) and BPE (red) models converge to a similar efficiency frontier at scale, a larger FLOPs penalty persists between MDM byte (green) and its BPE counterparts (yellow). **Data (Right):** When plotted against the volume of raw training data, AR models are practically overlaid, whereas MDM models retain a persistent performance gap even as total data volume increases.

deficit inherent to the byte-level objective when parameter count and data volume are held constant, independent of compute-balancing.

Following Nie et al. (2025), to determine the conditional log-likelihood of a sequence $x_0$ for downstream tasks, we employ the chain rule decomposition $\log p_\theta(x_0|\text{prompt}) = \sum_{i=0}^{L-1} \log p_\theta(x_0^{(i)}|\text{prompt}, x_0^{(<i)}, m)$. This allows for a direct comparison between the autoregressive models and the naturally bidirectional masked diffusion models on a unified sequential likelihood basis.

## 5. Quantifying the Scaling Overhead

We begin with an empirical investigation into the computational investment required for a byte model to reach performance parity with a BPE counterpart, following the compute-matched protocol defined in Section 4.

### 5.1. Scaling Trajectories and BPB Convergence

Figure 2 reveals the divergence in how modeling objectives navigate the transition from compressed to raw byte inputs.

**Autoregressive models achieve near-parity at scale.** As training compute increases, the performance gap between AR byte (purple) and AR BPE (red) models narrows. When evaluated against training FLOPs, byte-level models initially lag behind BPE baselines, but the trajectories begin to converge as they approach the $1e22$ FLOPs regime. This behavior indicates that for causal objectives, compute is an effective substitute for pre-computed discretization.

**Masked diffusion models exhibit an efficiency offset.** In contrast, MDM byte models (green) suffer a substantial performance penalty compared to MDM BPE (yellow). Although MDM byte models show a steeper learning trajectory, they remain significantly less efficient than their BPE

baselines across the studied scales. This persistent vertical offset in BPB indicates that the order-agnostic objective faces a fundamental structural difficulty in modeling granular sequences without the guidance of pre-composed semantic units.

The right side of Figure 2 highlights a similar disparity in data utilization. In the AR setting, BPE and byte models show comparable training data sample efficiency, with byte models frequently outperforming their BPE counterparts at higher compute budgets. Conversely, while the downward trend of byte MDM curves suggests room for future improvement, the persistent performance offset indicates a practical sample inefficiency: to match the perplexity of a BPE counterpart, a byte MDM requires much larger data volumes.

**Downstream Task Performance** To verify that these BPB differences translate to semantic capabilities, we evaluate our models on standard zero-shot reasoning benchmarks. Following the protocols defined in Section 4, we compare the 717M BPE baseline against its compute match (180M byte at the $F \approx 2 \times 10^{20}$ FLOPs budget) and its capacity match (717M byte).

Results in Table 1 demonstrate an advantage for AR modeling and for BPE representations. Consistent with the perplexity results, AR models outperform MDM models, with BPE baselines generally surpassing their byte equivalents. Notably, the AR byte model demonstrates an ability to narrow the gap to its BPE counterpart as we increase capacity from the compute-matched (180M) to the parameter-matched (717M) setting. In contrast, the MDM byte model stagnates. Despite receiving the same increase in capacity that benefited the AR model, MDM byte performance fluctuates, improving in some tasks while degrading in others, resulting in a consistently low average that fails to show meaningful scaling at these compute budgets.

| Model | Params | ARC-E | BoolQ | HellaS | OBQA | PIQA | RACE | SIQA | Avg |
|---|---|---|---|---|---|---|---|---|---|
| *Baseline ($D = 45 \times 10^9$ BPE tokens)* | | | | | | | | | |
| AR BPE | 717M | 49.49 | 59.45 | 41.93 | 36.40 | 67.08 | 30.91 | 38.33 | **46.23** |
| MDM BPE | 717M | 35.69 | 59.14 | 31.93 | 32.40 | 59.58 | 29.67 | 36.49 | 40.70 |
| *Compute Match ($D = 188 \times 10^9$ byte tokens, $N$ =180M)* | | | | | | | | | |
| AR byte | 180M | 43.01 | 54.62 | 35.74 | 30.80 | 63.11 | 28.42 | 36.28 | 41.71 |
| MDM byte | 180M | 26.68 | 60.64 | 29.03 | 31.00 | 55.28 | 27.85 | 33.62 | 37.73 |
| *Capacity Match ($D = 188 \times 10^9$ byte tokens)* | | | | | | | | | |
| AR byte | 717M | 47.60 | 56.29 | 39.29 | 33.00 | 66.32 | 30.14 | 37.00 | 44.23 |
| MDM byte | 717M | 30.60 | 44.86 | 30.33 | 31.60 | 56.75 | 27.18 | 36.03 | 36.76 |

*Table 1.* Comparison of byte vs. BPE models. AR byte models remain relatively competitive with BPE baselines, whereas MDM byte models significantly underperform, even when parameter counts are matched. (MDM likelihoods computed via chain rule decomposition). Last column shows the average of zero-shot accuracy scores across tasks.

At this scale, we found that all models struggle with complex tasks. We evaluated HumanEval (Chen et al., 2021), MBPP (Austin et al., 2021), and BBH (Srivastava et al., 2022; Suzgun et al., 2022), but observed trivial performance (near-zero or random chance). We omit these numbers to focus on BPB and simpler tasks where meaningful signals were present.

### 5.2. Quantifying the Performance-Compute Frontier

Rather than a static penalty, the representational overhead of byte modeling is a dynamic value that evolves with scale. We characterize this behavior following the scaling laws protocol established by Hoffmann et al. (2022).

**Divergent Scaling Trajectories** As visualized in Figure 3, the horizontal distance between BPE and byte minima represents the additional training compute required for a byte-level model to match its BPE counterpart. Our results demonstrate that the compute penalty of bytes is a dynamic efficiency ratio governed by the modeling objective. For AR models, the compute gap narrows quickly within the studied compute scales: at BPB= 1.0, the gap is $\approx 7.9\times$ ($2.3e20$ for byte vs. $2.9e19$ for BPE), and at BPB= 0.8 this gap shrinks to $\approx 2.3\times$ ($2.7e21$ for byte vs. $1.2e21$).

MDM byte models suffer a far greater penalty. At BPB= 1.2 the compute gap is $\approx 34.3\times$ ($1.2e21$ for bytes vs. $3.5e19$ for BPE). Scaling projections indicate that at BPB= 1.0, the gap remains as high as $\approx 20.5\times$ ($9.0e21$ vs. $4.4e20$).

**IsoFLOPs Curvature and Extrapolation** By fitting a power law to these optimal points, we predict that AR byte and BPE models will reach parity at approximately $F \approx 1.3e22$ FLOPs. For MDM, the steeper requirement suggests parity may only be achievable at extreme budgets of $F \approx 4.2e26$ FLOPs.

We then characterize the shifting performance gap between byte and BPE models across scale. A power law between the BPB ratio and compute budget quantifies the rate at which the byte-modeling penalty decays as a function of total training FLOPs. The derivation is shared in Appendix C. The results show that **cross scale, the penalty of byte modeling is worse for MDM than for AR.**

This persistent difference suggests that the order-agnostic nature of the diffusion objective interacts poorly with granular byte-level representations, a structural mismatch we investigate mechanistically in the following sections.

## 6. Emergent Representation and Implicit Segmentation

The narrowing of the scaling overhead for autoregressive models suggests they eventually develop internal mechanisms that approximate the advantages of a pre-computed tokenizer.

### 6.1. Entropy as a Proxy for Segmentation

To investigate this, we analyze the predictive entropy of trained AR byte models. We find that the predictive uncertainty is non-uniform, exhibiting high entropy at the start of frequent sub-word units and low entropy for predictable byte transitions within those units.

Figure 4 illustrates this alignment: regions of high entropy (dark red) track the initial (non-space) bytes of BPE tokens. The black-outlined boxes denote the first non-space byte of the corresponding BPE token from the Llama 2 tokenizer.

To quantify this alignment between predictive uncertainty and the structural units constructed by BPE, we frame the alignment as a binary classification task, using scalar entropy to predict BPE start boundaries. The resulting ROC AUC of 0.829 demonstrates a strong statistical alignment with the structures established by a BPE tokenizer.

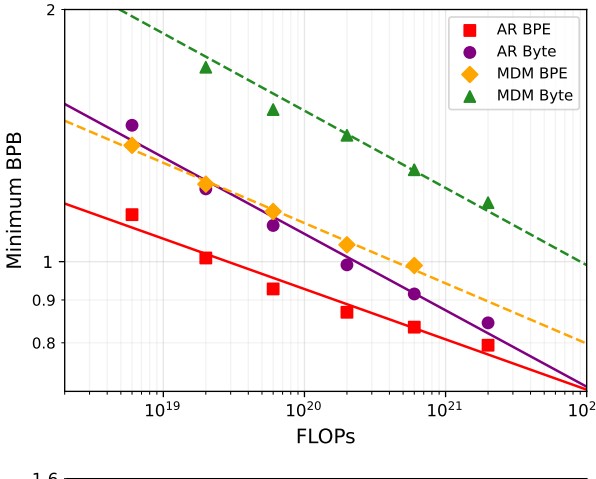

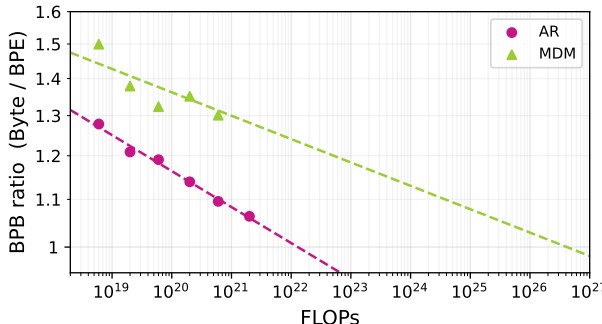

Figure 3. **(Top)** Extrapolated isoFLOPs minima are plotted against training FLOPs and fit to a power law. **(Bottom)** The BPB ratio is fit to a power law, showing that the gaps also close at different rates: byte modeling scales better in AR.

### 6.2. Implicit Segmentation as an Objective-Dependent Requirement

The behavioral alignment between predictive entropy and sub-word boundaries suggests that the predictive efficiency of autoregressive byte models is supported by their sensitivity to local statistical dependencies. The sharp transition in entropy at these boundaries indicates that the model successfully leverages its stable causal history to resolve high-frequency transitions within frequent byte patterns.

We focus this analysis on AR models as they provide a stable representational baseline. Obtaining a unified entropy measure for MDM is more complicated due to the objective's stochastic nature and combinatorial masking states. If the causal guarantee of a contiguous prefix is a primary driver of compute-efficiency in learning to model data, then the stochastic masking inherent in byte MDM may fundamentally hinder the development of this structural wordpiece-like awareness. To test this, we move from observing emergent structures to a series of controlled experiments designed to scramble this context and quantify the resulting fragility of the byte representation.

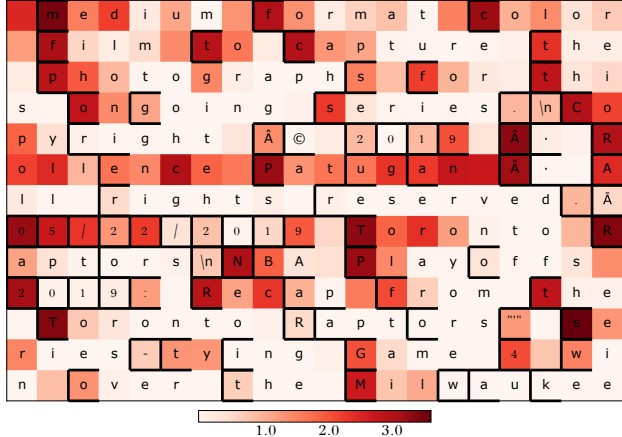

Figure 4. Scalar predictive entropy of an AR Byte model is visualized for a sample text. Regions of high entropy (dark red) align with the first non-space byte of corresponding BPE tokens (black outlines), achieving an ROC AUC of $0.829$ for predicting BPE start boundaries.

## 7. Mechanism of Failure: Context Fragility

We hypothesize that the performance degradation of byte-level diffusion stems from the fragility of byte sequences under context corruption. While a BPE token often corresponds to a subword unit with intrinsic semantic meaning, a single byte is semantically vacuous in isolation. Consequently, byte-based sequences are uniquely susceptible to losing informational value when their sequential integrity is compromised.

**Experimental Setup: Permutation as Corruption** To test this, we simulate the context destruction of out-of-order generation within a controlled AR framework. We apply a static permutation ($\pi$) to sequences where both the token embedding and their corresponding position ID embeddings are permuted together (Pannatier et al., 2024). We compare three permutation strategies (visualized in Figure 6): (1) **global random**: all token positions are randomly shuffled, destroying local and global structure; (2) **inter-block-k random**: blocks of size $k$ bytes are shuffled while internal byte order is preserved, destroying global order while preserving local contiguity; (3) **intra-block-k random**: bytes within each block of size $k$ are shuffled, destroying local order while preserving global order to isolate the role of a global causal sequence.

To decouple the model's predictive capabilities from the inherent randomness of the shuffled data, we utilize average loss in compressibility under the DEFLATE algorithm (Deutsch, 1996) as a model-free proxy for data regularity. Additional model-free baselines for data regularity that support these trends are provided in Appendix E.

Training dynamics in Figure 6 reveal that the model's perfor-

| π **Strategy** | **Visualization** | **Compr.** |
|---|---|---|
| Original BPE | `[diffusion][_models]` | – |
| Global BPE ■ | `[_models][diffusion]` | -7% |
| Original | `diffusion_models` | – |
| Global ■ | `_musisdeofnodlif` | -19% |
| Inter-4 ■ | `n_mo diff dels usio` | -14% |
| Inter-8 ■ | `n_models diffusio` | -9% |
| Intra-8 ■ | `ifdfsiuo noeld_ms` | -16% |

*Figure 5.* Corruption strategies on the string `diffusion_models`. Average loss in compressibility (%) under the DEFLATE algorithm serves as a model-free proxy for probabilistic structure.

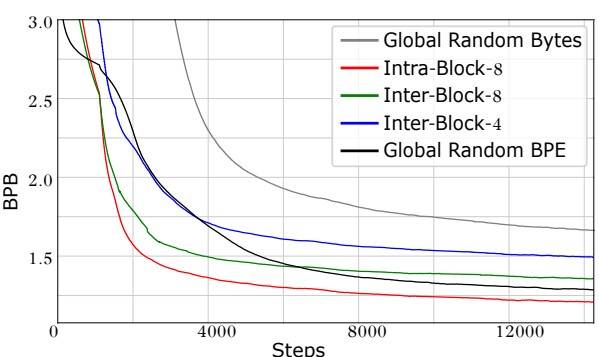

*Figure 6.* We compare the impact of different permutation strategies on training dynamics. Global random permutation (gray) degrades byte modeling performance most. While local contiguity aids recovery (Inter-Block, green/blue), preserving global causal order (Intra-Block, red) yields the best results, outperforming permuted BPE baseline. This highlights the unique robustness provided by causal history.

mance is not a simple reflection of data entropy, but is driven by the interaction between objective and representation.

**Byte models are uniquely fragile to context loss**. The AR byte model suffers a much sharper performance drop under global random permutation compared to AR BPE. This is confirmed by our DEFLATE baseline, which shows that raw byte streams lose significantly more structural information ($-19\%$) than BPE ($-7\%$) under identical corruption. This suggests that BPE tokens act as robust, pre-packaged semantic units that remain learnable even when global context is fragmented.

**Local contiguity is a helpful inductive bias**. Under inter-block-k permutation (preserving local chunks), byte models partially recover. This tracks the compressibility trend, where preserving local byte-chunks allows both the compressor and the model to learn frequent sub-word patterns.

**Causal context compensates for local noise**. Interestingly, we discover an inversion between data compressibility and model performance under intra-block-k permuted sequences. Under this strategy (preserving global order), the model achieves its best recovery, outperforming the inter-block-8 setting and even the globally-permuted BPE baseline. This occurs even though the data is significantly less compressible ($-16\%$ loss under DEFLATE) than the inter-block-k settings. This provides strong evidence that stable causal history is a powerful inductive bias for modeling, as the model extracts value from global order that simpler statistical compressors cannot.

**Implication for MDM** These findings identify a fundamental structural mismatch in byte-level diffusion. While AR models may naturally resolve granular dependencies through their causal objective, the byte MDM objective is doubly destructive: it operates on granular units (no encapsulation) while simultaneously scattering the context required to resolve them (no causal history). Deprived of both the pre-composed wordpiece units of BPE and the stable causal anchors of AR, the diffusion model must resolve semantic dependencies across a combinatorial landscape of possible orderings. Our results suggest that this context fragility drives the disparate scaling behavior observed across paradigms, and implies that future byte-level MDMs may require specific architectural adaptations to maintain comparable performance to the alternatives under these compute scales.

**Vocabulary Sensitivity** Finally, we investigate whether these trends are sensitive to the specific subword vocabulary used. We evaluate GPT-2 ($V \approx 50k$) and Llama-3 ($V \approx 128k$) tokenizers and find that while larger vocabularies offer higher compression, the optimal vocabulary size for masked diffusion depends on compute budget. Results in Appendix G show that the byte-BPE gap persists across vocabulary scales.

## 8. Conclusion

In conclusion, we have quantified the scaling overhead of byte-level representations relative to subword tokenization, revealing that the computational penalty is dependent on modeling objective. Specifically: cross-scale, the compute penalty of byte modeling is more severe for MDM than for AR. While autoregressive models effectively amortize these raw-modeling costs at scale, byte-level MDMs suffer a performance offset. This behavior likely stems from context fragility; our mechanistic investigations suggest that the MDM objective shatters the local contiguity and global context required to resolve sub-word semantics from granular byte streams. Future research into tokenizer-free modeling should explore architectural adaptations, such as hierarchical masking, optimal vocabularies, or hybrid backbones, to maintain efficient and sustainable scaling trajectories.

## Impact Statement

Our findings suggest several avenues for future investigation into the interaction between data representation and modeling objectives. Compute-matched baselines identify important work to be done in designing data representations and training methods that align with specific modeling objectives. The compute-controlled pre-training required for this study involves significant computational resources and associated energy consumption, but our findings identify important structural opportunities and inefficiencies that provide the foundation for investigating more compute-efficient architectures. We believe our contributions will reduce the long-term environmental impact of training large-scale, modality-agnostic models.

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

# A. Discretization of Continuous-Time Diffusion

We follow the continuous-time discrete diffusion framework presented by Shi et al. (Shi et al., 2024) (MD4), which generalizes the discrete diffusion models of Austin et al. (Austin et al., 2023) (D3PM). Here we detail the forward process, training objective, and sampling strategy used in our experiments.

**Forward Process (Corruption).**    We consider a sequence $x_0$ of length $L$ tokens, where each token $x_0^{(i)}$ belongs to a finite vocabulary comprised of the model vocabulary plus a special mask token $\mathcal{V} \cup \{[\texttt{MASK}]\}$. The forward process is characterized as a continuous-time Markov chain over time interval $t \in [0, 1]$ applied independently to each token.

At any time $t$, the marginal distribution of the noisy sequence $x_t$ factorizes as $q(x_t|x_0) = \prod_{i=1}^{L} q(x_t^{(i)}|x_0^{(i)})$, where:

$$q(x_t^{(i)}|x_0^{(i)}) = \begin{cases} \alpha_t & \text{if } x_t^{(i)} = x_0^{(i)} \\ 1 - \alpha_t & \text{if } x_t^{(i)} = [\texttt{MASK}] \\ 0 & \text{otherwise} \end{cases} \tag{1}$$

Here, $\alpha_t$ is a monotonically decreasing masking schedule from $\alpha_0 \approx 1$ (no masking) to $\alpha_1 \approx 0$ (fully masked).

**Training Objective (Continuous-Time ELBO)**    The generative process is parameterized by a neural network $p_\theta(x_0|x_t, t)$ trained to predict the original uncorrupted sequence $x_0$ from the noisy state $x_t$. Following Shi et al. (Shi et al., 2024), we minimize the simplified continuous-time variational lower bound (ELBO), which reduces to a weighted cross-entropy over the masked tokens:

$$\mathcal{L}_{\text{MDM}}(\theta) = \mathbb{E}_{t \sim \mathcal{U}(0,1)} \left[ \underbrace{\frac{\alpha_t'}{1 - \alpha_t}}_{w(t)} \sum_{i \in \{i : x_t^{(i)} = [\texttt{MASK}]\}} \log p_\theta(x_0^{(i)}|x_t, t) \right] \tag{2}$$

Note that because $\alpha_t$ is decreasing over $t$, the derivative $\alpha_t'$ is negative, making $w(t)$ negative, which offsets the negative value produced by the $\log p_\theta$.

**Sampling: Reverse Process (Denoising).**    During inference, we simulate the reverse process by discretizing time into $T$ steps: $1 = t_T > \cdots > t_0 = 0$. At each step $t \to s$ (where $s < t$), we update the sequence based on the model's prediction. Tokens that are already unmasked are kept fixed. For tokens that are still masked at time $t$, we sample their value at the next step $s$ according to:

$$x_s^{(i)} \sim \begin{cases} \text{Cat}(\frac{\alpha_s - \alpha_t}{1 - \alpha_t} \sigma(f_\theta(x_t, t))^{(i)} + \frac{1 - \alpha_s}{1 - \alpha_t} e_m) & \text{if } x_t^{(i)} = [\texttt{MASK}] \\ x_t^{(i)} & \text{else} \end{cases} \tag{3}$$

**Implementation and Schedules.**    We approximate the expectation over time by sampling $t \sim \mathcal{U}(0, 1)$ for each batch. We use a linear masking schedule during training and cosine schedule during evaluation and sampling.

# B. FLOPs Computation

We calculate the forward pass FLOPs for our Transformer architecture based on the specific operations of our decoder-only backbone. We largely follow the methodology of Hoffmann et al. (2022), with modifications to account for the SwiGLU activation functions used in the Transformer architecture.

For a single forward pass, the FLOPs are computed as follows:

- **Embeddings**: $2 \times L \times V \times d_{\text{model}}$

- **Attention Layer** (per layer):

    - **QKV Projections**: $3 \times 2 \times L \times d_{\text{model}} \times (n_{\text{heads}} \times d_{\text{head}})$
    - **Attention Logit Calculation** ($QK^T$): $2 \times L^2 \times (n_{\text{heads}} \times d_{\text{head}})$
    - **Attention Softmax Weighting**: $2 \times L^2 \times (n_{\text{heads}} \times d_{\text{head}})$

– **Output Projection**: $2 \times L \times (n_{\text{heads}} \times d_{\text{head}}) \times d_{\text{model}}$

- **MLP Block (SwiGLU)** (per layer):
    – **Up-Projections and Gating**: $2 \times (2 \times L \times d_{\text{model}} \times d_{\text{ff}})$
    – **Down-Projection**: $2 \times L \times d_{\text{ff}} \times d_{\text{model}}$

- **Language Modeling Head**: $2 \times L \times d_{\text{model}} \times V$

The total training compute ($F$) is then calculated by multiplying the per-step forward pass FLOPs by a factor of 3 to account for the backward pass and gradient computation, and then scaling by the total number of tokens processed.

## C. Power Laws for the BPB Ratio

Given the relationship between training budget $F$ and BPB score that is linear in log-log space with slope $s$ and intercept $b$:

$$\log(BPB) = s \times \log(F) + b \tag{4}$$

The performance ratio between byte-level and BPE-level models can be derived as as a function of the slope ($\Delta_s$) and offset ($\Delta_b$) differences of the fit power-laws:

$$\log(BPB_{\text{ratio}}) = \log\left[\frac{exp(b_{\text{byte}}) \times F^{s_{\text{byte}}}}{exp(b_{\text{BPE}}) \times F^{s_{\text{BPE}}}}\right] \tag{5}$$

$$= \log\left[exp(\Delta_b) \times F^{\Delta_s}\right] \tag{6}$$

$$= \Delta_b + (\Delta_s)\log(F) \tag{7}$$

## D. Isolating the Effects of Context Length

To decisively decouple representational complexity from context length, we run controlled stress tests training BPE and Byte models at fixed, identical sequence lengths across multiple compute budgets. Figure 7 illustrates that even under the same context length, thus incurring the same quadratic scaling, byte models underperform their BPE counterparts at lower compute budgets, with the gap particularly pronounced in masked diffusion models.

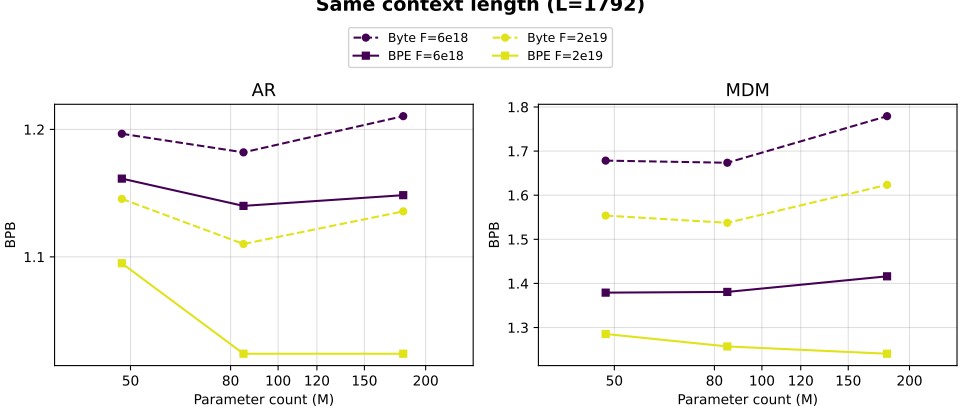

*Figure 7.* Even with the same context length, byte models (dashed lines) underperform BPE counterparts (solid lines) at the same compute budget (same color).

## E. More Proxies for Data Compressibility

DEFLATE (Deutsch, 1996), which combines LZ77 prefix matching with Huffman coding, lets us quantify the statistical regularities and repetitive patterns preserved under various permutations, providing a quantitative baseline for the entropy offset introduced by context corruption. We also measure compressibility under other algorithms with different advantages

| $\pi$ **Strategy** | DEFLATE (v9) | LZMA (v9) | Zstd (v22) | PPMd (O8) |
|---|---|---|---|---|
| Global BPE | 6.9% | 6.0% | 6.6% | 4.8% |
| Global Byte | 19.3% | 18.0% | 14.9% | 24.2% |
| Inter-4 | 14.0% | 14.4% | 13.3% | 15.9% |
| Inter-8 | 9.0% | 9.5% | 8.7% | 10.0% |
| Intra-8 | 16.4% | 14.9% | 13.1% | 20.3% |

*Table 2.* Average loss in compressibility (%) under various compression algorithms serve as model-free proxies for data regularity.

over length and data format: Zstd (Collet & Kucherawy, 2021), LZMA (Pavlov, 2024) (high-compression algorithm optimized for long-range repetitions), and PPMd (Shkarin, 2001) Order-8 (a Markovian predictor closer to a statistical language model and particularly good at compressing text data). Across all four compression engines, the trend remains robust: the destruction of local contiguity in Intra-8 and global shuffles produces the most significant increase in data entropy.

## F. The Failure of Naive Span Masking

A natural follow-up to the results of the block-based permutation experiments in Section 7 is to question whether enforcing local contiguity during training can improve Byte MDM performance. The standard MDM forward process destroys local context by masking tokens independently. We hypothesize that by modifying the forward process to mask contiguous blocks of bytes (spans), we might allow the model to leverage unmasked neighbors to better resolve semantic ambiguity.

**Methodology.** We implement a BPE-based span masking strategy. Instead of sampling the mask state for each token $x^{(i)}$ independently, we partition the sequence indices into disjoint blocks $\{B_1, ... B_K\}$ where each block corresponds to continuous bytes defined by a BPE tokenizer. This span-based forward process factorizes over blocks instead of individual tokens. For a block $B_k$, the transition probability is defined as:

$$q(\mathbf{x}_t^{(B_k)}|x_0^{(B_k)}) = \begin{cases} \alpha_t & \text{if } \forall i \in B_k, x_t^{(i)} = x_0^{(i)} \\ (1-\alpha_t) & \text{if } \forall i \in B_k, x_t^{(i)} = \texttt{[MASK]} \\ 0 & \text{otherwise} \end{cases} \tag{8}$$

This formulation ensures strict coupling: with probability $\alpha_t$, the entire BPE-based byte chunk is preserved, and with probability $1 - \alpha_t$ the entire chunk is masked. The full sequence probability, as before, is simply the product over all blocks: $q(x_t|x_0) = \prod_{k=1}^{K} q(\mathbf{x}_t^{(B_k)}|\mathbf{x}_0^{(B_k)})$.

**Results.** Figure 8 presents the validation BPB curves for 180M parameter MDM models. We observe a strict performance hierarchy: BPE MDM performs better than byte MDM which performs better than a BPE-based masking strategy over byte modeling.

In this experiment, restoring local context via span masking hurt performance. This, at first glance, seems counterintuitive when the restoring of local context in the permutation experiments (Section 7) helped. But the difference lies in the prediction task. BPE tokenization compresses the output space: repeatedly predicting a $\approx$ 4-byte word via a 1-in-32k classification task. Naive span masking keeps the bytes independent, meaning the model must implicitly resolve a 1-in-$256^N$ classification task for spans of size $k$. In the permutation experiment, the model still predicts the next token $x_t$ given a (jumbled) history. In span masking, the model must jointly reconstruct a block of multiple contiguously missing bytes.

## G. An Upper Limit to Vocabulary Efficiency

To test if larger vocabularies yield further gains, we extended our sweep to include GPT-2 ($V \approx 50k$) and Llama-3 ($V \approx 128k$) tokenizers. We find that BPE MDM models consistently outperform byte counterparts regardless of vocabulary size. Our results indicate that the optimal vocabulary size for MDM is not static; it requires balancing semantic density against the increased classification difficulty of massive vocabularies.

Over the compute settings sizes described in Section 4 ($N = 180M, 717M$), we find, consistent with the main results of this paper, that BPE MDM models consistently outperform their Byte counterparts regardless of vocabulary size.

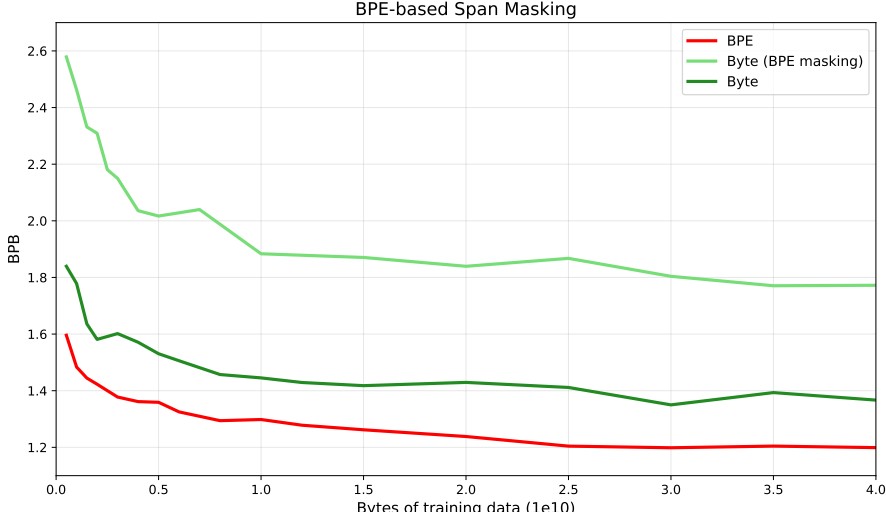

*Figure 8.* **Span Masking Performance.** Validation BPB for MDM models trained with BPE tokens (red), byte tokens with byte-granular masking (dark green), and byte tokens with BPE-based span masking (light green). Contrary to the intuition that preserving context helps, larger spans monotonically degrade performance.

| Tokenizer | Vocab ($V$) | Avg. Bytes |
|---|---|---|
| Byte | 256 | 1.00 |
| Llama-2 | 32,000 | 3.74 |
| GPT-2 | 50,257 | 4.16 |
| Llama-3 | 128,000 | 4.36 |

*Table 3.* Tokenizer vocabulary size and compression rates.

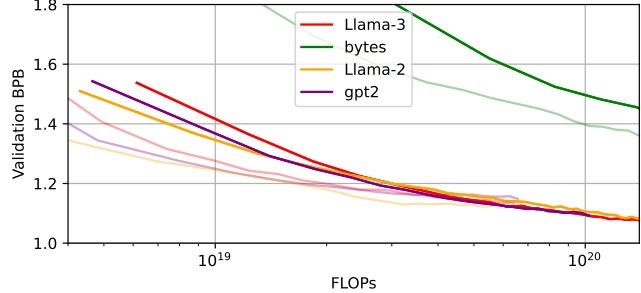

*Figure 9.* Iso-FLOPs curves for 180M and 717M models. Larger vocabularies offer higher compression.

As shown in Figure 9, we observe that the best-performing vocabulary size depends on model capacity and compute budget. Observe, for example, that for the smaller 180M model, the Llama-2 tokenizer ($V = 32k$) fares the best. But as the FLOPs budget increases to $10^{20}$ for the larger 717M model, the larger BPE tokenizers (GPT-2, Llama-3) close the gap, even overtaking the smaller-vocabulary Llama-2 tokenizer baseline. These results indicate that optimal vocabulary size for masked diffusion is not static; it requires balancing the benefits of semantic density against the token sparsity and classification difficulty of massive vocabularies.

