# OpenReview forum: "The Efficiency Gap in Byte Modeling"
_ICML.cc/2026/Conference — ICML 2026 regular_

### Official Review · Reviewer_uYkH · 2026-02-16

**Soundness:** 3
**Presentation:** 2
**Significance:** 2
**Originality:** 2
**Overall Recommendation:** 4
**Confidence:** 4

**Summary:**

This paper conducts a systematic comparison across the cross-product of modeling objectives and tokenization strategies, namely {AR, MDM} × {Byte, BPE}, under a compute-matched scaling framework. The goal is to disentangle how data representation (raw bytes vs. subword tokens) interacts with generation objective (autoregressive vs. masked diffusion).

The results reveal a striking objective-dependent divergence. For autoregressive (AR) models, the performance gap between Byte and BPE representations narrows as scale increases. The authors attribute this convergence to emergent segmentation: AR byte-level models implicitly rediscover subword-like structures during training. This is evidenced by the alignment between predictive entropy peaks and BPE token boundaries, suggesting that stable causal history enables the model to internalize tokenization structure and amortize the cost of operating on raw bytes.

In contrast, masked diffusion models (MDMs) exhibit a persistent efficiency collapse when trained on byte-level inputs. Unlike AR models, MDMs fail to recover comparable performance to their BPE counterparts even at higher compute budgets. The authors argue that this failure stems from context fragility: the masking objective disrupts both local contiguity and global causal order, preventing the formation of stable subword-level abstractions. As a result, byte-level MDMs must resolve semantic dependencies from low-signal units under corrupted context, leading to substantial inefficiency.

Overall, the study demonstrates that the viability of byte-level modeling is strongly objective-dependent: while AR models can internally reconstruct segmentation structure at scale, MDMs struggle due to structural incompatibility between granular representations and order-agnostic masking.

**Compliance With Llm Reviewing Policy:**

Affirmed.

**Key Questions For Authors:**

If the MDM is modified to explicitly preserve local contiguity—e.g.,
through hierarchical/block-wise masking or span-based corruption that
maintains subword-level coherence—does the byte-level efficiency
collapse still persist?

**Limitations:**

The paper would benefit from a clearer and more explicit discussion of its limitations, particularly regarding the restricted MDM configuration and the generality of the context fragility claim.

A limitation of the paper is that the context fragility hypothesis, while compelling, is not directly stress-tested within the MDM setting itself. The authors attribute the byte-level efficiency collapse under MDM to the destruction of local contiguity and causal structure, yet they do not perform ablation studies that explicitly vary the masking strategy within MDM to test this claim. In particular, experiments using locality-preserving masking (e.g., span-based or block-wise corruption), hierarchical diffusion, or progressively structured masking schedules are not explored. Without such controlled modifications, it remains unclear whether the observed inefficiency is fundamentally inherent to byte-level diffusion, or instead a consequence of the specific masking configuration employed. As a result, the generality of the conclusion regarding byte-level MDM may be overstated.

**Strengths And Weaknesses:**

This paper addresses a timely and important question at the intersection of tokenizer-free modeling and non-autoregressive generation. The experimental setup is rigorous, employing a compute-matched protocol across the full {AR, MDM} × {Byte, BPE} design, which enables a fair and systematic analysis of scaling behavior.

A central strength of the paper lies in its clear structural framing of the efficiency gap as objective-dependent rather than representation-dependent alone. Instead of merely reporting performance differences, the authors provide a compelling mechanistic account through the context fragility hypothesis and supporting permutation experiments. Most importantly, the paper offers a precise answer to the question, “Is byte-level modeling always efficient?”—demonstrating that tokenizer-free modeling is viable under autoregressive objectives due to emergent segmentation, but suffers a persistent efficiency collapse under masked diffusion. This reframing provides an important conceptual clarification for the broader tokenizer-free modeling debate and highlights the critical role of objective–representation compatibility.


The main limitation of the paper lies in the simplicity of the MDM configuration. The conclusion that byte-level modeling leads to an efficiency collapse under masked diffusion may be overly broad, given that the experiments rely on a relatively standard, minimally modified MDM setup. The study does not explore alternative diffusion variants—such as hierarchical masking, block diffusion, locality-preserving corruption schedules, hybrid AR–diffusion architectures, or architectures specifically designed for long byte sequences. As a result, it remains unclear whether the observed inefficiency reflects a fundamental incompatibility between byte representations and diffusion objectives, or merely the limitations of the specific MDM instantiation used in this work. Without a broader exploration of the design space, the generality of the negative conclusion for MDM may be overstated.

---

> ### Author Rebuttal · Authors · 2026-03-31
>
> We thank the reviewer for their positive assessment of our context fragility hypothesis and our compute-matched protocol.
>
> ## re: scope and architectural isolation
>
> The reviewer raises an excellent point regarding the standard configuration of our MDM baseline. We intentionally used a standard backbone to isolate fundamental objective-representation interactions without confounding architectural variables. Per your suggestion, we evaluate three variants:
> Linear attention backbones: to test if the gap is simply an artifact of quadratic attention, we replaced the backbone with linear attention (Katharopoulos et al., https://arxiv.org/abs/2006.16236) . While this allows for more training data within the same compute budget, the performance deficit across representations persists: the MDM linear-byte model (BPB=1.635) remains significantly behind its BPE counterpart (BPB=1.119).
>
> | Model Variant | 2.00E+19 FLOPs | 6.00E+19 FLOPs |
> | :--- | ---: | ---: |
> | **AR BPE** | 1.007 | 0.927 |
> | **MDM BPE** | 1.247 | 1.173 |
> | **AR Byte** | 1.258 | 1.129 |
> | **MDM Byte** | 1.813 | 1.617 |
> | **AR-Linear attention-BPE** | 0.995 | 0.912 |
> | **AR-Linear Attention Byte**| 1.075 | 1.072 |
> | **MDM-Linear attention** | 1.236 | 1.119 |
> | **MDM-Linear Attention Byte**| 1.799 | 1.635 |
>
> * Alternate masking schedules: evaluating a cosine masking schedule (Shi et al. https://arxiv.org/abs/2406.04329) showed slight absolute improvements, but the byte-BPE gap remains wide:
>
> | FLOPs Budget | MDM BPE | MDM Byte | MDM BPE-cosine | MDM Byte -cosine |
> | :---: | ---: | ---: | ---: | ---: |
> | **2.00E+19** | 1.247 | 1.813 | 1.231 | 1.792 |
> | **6.00E+19** | 1.173 | 1.617 | 1.169 | 1.547 |
>
> * Block diffusion: we evaluated block diffusion (Arriola et al. https://arxiv.org/abs/2503.09573) across multiple block sizes. As shown in the table below, byte models consistently underperform BPE counterparts across block sizes.
>
> | Block Size | FLOPs Budget | Schedule | MDM BPE | MDM Byte |
> | :---: | :---: | :--- | :--- | :--- |
> | **baseline** | 6.00E+19 | linear | 1.173 | 1.617 |
> | **2** | 6.00E+19 | linear | 1.176 | 1.619 |
> | **8** | 6.00E+19 | linear | 1.324 | 1.828 |
> | **16** | 6.00E+19 | linear | 1.340 | 1.862 |
> | **32** | 6.00E+19 | linear | 1.322 | 1.816 |
> | **baseline** | 6.00E+19 | cosine | 1.169 | 1.547 |
> | **2** | 6.00E+19 | cosine | 0.923 | 1.528 |
> | **8** | 6.00E+19 | cosine | 1.189 | 1.725 |
> | **16** | 6.00E+19 | cosine | 0.988 | 1.757 |
> | **32** | 6.00E+19 | cosine | 1.057 | 1.715 |
>
>
> * Span-based masking: Our results show that span-based masking introduces a destructive trade-off between local contiguity and informational distance: multi-token spans move the nearest available (i.e. unmasked) information further away from the units being predicted. This forces the model to resolve dependencies across even larger information gaps, effectively exacerbating the broken-context problem. Our 180M MDM study confirms that masking whole BPE-sized byte spans actually hurts performance compared to 1-gram masking: [see results here](https://imgur.com/a/qaP3qy3)
>
>
> Our study establishes differences in scaling behavior when different modeling objectives and representations interact, providing the necessary empirical motivation for the specialized architectural and training adaptations suggested by the reviewer. We believe our findings set an important basis for future tokenizer-free research, and we hope our responses reinforce the value of our contributions.
>
> Supporting media:
> *  [span-based masking](https://imgur.com/a/qaP3qy3)

---

> > ### Author Rebuttal · Reviewer_uYkH · 2026-04-03
> >
> > The authors have fully resolved my concerns. I think this paper discusses the limitations of MDM objective. I will keep my score being 4.

---

### Official Review · Reviewer_KNsv · 2026-02-21

**Soundness:** 3
**Presentation:** 3
**Significance:** 2
**Originality:** 3
**Overall Recommendation:** 4
**Confidence:** 3

**Summary:**

The paper provides a systematic study of the "efficiency gap" incurred when bypassing BPE for raw bytes across different modeling objectives. Through a compute-matched scaling study (up to 1.2B parameters), the authors reveal a structural dichotomy: while AR models on bytes eventually "rediscover" subword structures at scale, byte-level MDMs suffer a persistent and significantly steeper efficiency penalty.

**Compliance With Llm Reviewing Policy:**

Affirmed.

**Final Justification:**

The rebuttal addressed my main concerns.

**Key Questions For Authors:**

1. How can you be certain the "efficiency collapse" in MDM is a failure of the byte representation specifically, rather than the well-documented difficulty of Transformer backbones to manage dependencies over 4x longer sequences?

**Limitations:**

1. The paper doesn't exclude the impact of sequence length on different objectives.

**Strengths And Weaknesses:**

**Strengths**
1. Rigorous Scaling Methodology: The authors employ a compute-matched evaluation protocol, ensuring comparisons are grounded in total FLOPs expended rather than just parameter counts.
2. Mechanistic Insights: The paper goes beyond simple observation by providing a structural analysis of why MDMs fail in the byte regime.

**Weaknesses**

1. Incomplete Isolation of Sequence Length Effects: The paper compares byte sequences of length 8192 with BPE sequences of length 1792 to normalize "information content". However, by using sequences over four times longer for bytes, the model faces a much harder optimization task regardless of the token type.
2. The authors attribute the MDM failure to the objective "shattering" local contiguity. However, if we view a sentence as a "package" of tokens, and a token is just a "package" of bytes, the objective of recovering a missing piece is the same regardless of the zoom level. MDM on bytes might even be easier because predicting tokens requires selecting from a massive vocabulary, while predicting bytes only require choosing from 256 options. Also, a sentence contains more information than a token, recovering missing pieces(tokens) from a sentence is intuitively harder than recovering missing pieces(bytes) from a token.

---

> ### Author Rebuttal · Authors · 2026-03-31
>
> We thank the reviewer for their thoughtful feedback and for recognizing the rigor of our compute-matched evaluation protocol. We appreciate the highlight of our mechanistic insights, particularly the structural analysis of why MDMs struggle in the byte regime compared to AR counterparts.
>
> ## re: sequence length isolation
>
> We appreciate the observation regarding context length as a potential confounding factor. To decouple the effects of increased context length and representational complexity, we conducted a targeted stress test: we trained BPE and Byte models of three sizes (48M, 85M, 180M) at identical context length of  $L=1792$ across two compute budgets ($F=6e18, F=2e19$). As shown in our results [here](https://imgur.com/a/QYxpkY6), even when the sequence length is equalized, the gap from BPE to Byte models persist. Notably, reducing the context length allows the model to see much more data within the same compute budget, but even with this aid the byte models still underperform, confirming that the compute penalty is representational, not just a matter of attention costs. Please see our Linear Attention results in the response to reviewer uYkH for further isolation of this effect.
>
> ## re: information density and classification difficulty
>
> The reviewer correctly identifies the disparity in vocabulary size (256 vs. 32k) as a point of task-complexity disparity. While the classification head is smaller for a byte model, the generative task is more complex: to resolve a single 4-byte semantic unit (a subword), a byte model must navigate a combinatorial search space of
> $256^4 ~ 4.3*10^9$ possibilities.
> In contrast, a BPE model selects that same unit in a single step from a 32k vocabulary.
>
> Individual bytes carry minimal semantic value in isolation, whereas BPE tokens act as a pre-computed compression layer structured around the probabilistic patterns of language. Using the bits-per-byte (BPB) metric allows our results to decouple predictive performance from discretization. As our results show, the MDM objective is uniquely destructive for bytes because it shatters the local contiguity needed to resolve these granular dependencies while denying the model the stable causal history leveraged by AR models.
>
>
> By demonstrating that the challenge of MDM byte modeling is a matter of semantic cohesion rather than simply context length, our work provides empirical evidence that the ``zoom level’’ of the tokenizer fundamentally dictates scaling efficiency. We believe these mechanistic insights offer a vital signal to the community regarding the design of future tokenizer-free architectures.
>
> Supporting media:
> * [controlled context length experiment](https://imgur.com/a/QYxpkY6)

---

> > ### Author Rebuttal · Reviewer_KNsv · 2026-04-02
> >
> > The authors have fully resolved my concerns. I think this paper indeed discovers the limitation of MDM objective. I would raise my score to 4.

---

### Official Review · Reviewer_7a9z · 2026-03-28

**Soundness:** 2
**Presentation:** 3
**Significance:** 2
**Originality:** 3
**Overall Recommendation:** 3
**Confidence:** 2

**Summary:**

The authors study two key design decisions in language modeling: tokenization strategy and modeling objectives. Specifically, they investigate the scaling properties (as measured by bits-per-byte) of auto-regressive and diffusion models, using BPE and byte-level tokenization. The paper finds that while autoregressive models show near-parity between byte and BPE tokenization as compute and training data scales up, the gap between the two tokenization methods remains large for diffusion models.

The authors argue that this difference is due to the masking structure of MDMs losing information about local structure and context. To support this, they cite the close alignment between predictive entropy and BPE boundaries, and conduct experiments using different methods of permuting language sequences to show that byte-level tokenization is particularly susceptible to losing information when causal structure is lost.

**Compliance With Llm Reviewing Policy:**

Affirmed.

**Final Justification:**

The authors have addressed my main concerns regarding section 5, and some of my questions about the results in section 7. However,  the empirical justification for their context fragility hypothesis still seems rather weak. Nevertheless I raised my score, given that the new experiments in the rebuttals provide better support for many of the authors' conclusions.

**Key Questions For Authors:**

See weaknesses section.

Additional questions:

1) In table 1, the drop for MDM Byte between compute match and capacity match seems largely dominated by a drop in BoolQ performance. Do the authors have any insight to why this might be the case?

2) Is there a similar plot to Figure 3 plotted against training data size? The AR plots look very similar in Figure 2 (Left) due to the presence of multiple lines for different model sizes, so plotting only the optimal model size could make the trends clearer.

3) Can the authors clarify what they mean by the MDM objective having "no encapsulation" or "no causal history" (Section 7)? From my understanding the MDM still sees positional embeddings that it can use to infer causal relations.

**Limitations:**

yes

**Strengths And Weaknesses:**

Strengths:

1) This work tackles important design decisions in language modeling, and addresses an interesting area of research
2) The authors perform comprehensive experiments on scaling these models (Fig 2)
3) Paper is generally clear and data is well-presented

Weaknesses:

1) While experiments in Fig 2. are comprehensive, they do not seem to back up the conclusions that 1) AR models achieve near-parity at scale, and 2) The (compute) efficiency gap between BPE and Byte is worse than in AR models. Looking at the compute-efficiency curves:

    * While the gap in absolute BPB does seem to decrease, the rate of BPB improvement as compute scales up also decreases sharply. For a given BPB level, it takes roughly 10x the compute to train an AR byte model vs an AR BPE model, and this 10x factor seems consistent as we scale up (looking at BPE=1.2, and BPE~0.9). Therefore it seems like there is a non-decreasing efficiency gap even for autoregressive models.

    * Moreover, the MDM plots suggest that the compute difference between byte and BPE encodings for the same BPB is also about 10x, not drastically different from the AR setting. This suggests that the compute scaling behavior of byte vs. BPE is rather similar in both the AR and MDM settings, contrary to what the authors claim.

2) It seems to me that the experiments in Section 7 are also flawed because the metric used is measuring log-probs, which is not just a function of the model or what it has learned, but also of the data. Specifically, the authors used different permutation methods to corrupt the training data for the model, which will introduce different amounts of randomness or entropy into the data. Because the same corrupted data is being used to measure BPB, the differences in entropy will show up as offsets in the log-probs (or BPB) in Fig. 6. Therefore, it is not clear that the differences in these plots are actually because of any differences in training dynamics of the model (as the authors claim), or if they are simply a property of the different datasets used.

3) The MDM plots for Figure 3 are linear fits using 2 data points, so the conclusions about parity for MDM are very uncertain. The information in this section also seems very similar to what was presented in 5.1; It would help if the authors could clarify the differences between the 2 sections.

4) Minor: some sections (such as section 6.2, in particular "leverages a stable causal history to resolve high-frequency transitions within frequent byte patterns") seem unclear. Consider rephrasing or explaining in more detail.

Nit: in line 95, "while byte models are… MDM masking" seems like a typo or incomplete sentence.

---

> ### Author Rebuttal · Authors · 2026-03-31
>
> We thank reviewer 7a9z for their detailed read of our work, and for the constructive feedback that allows us to clarify our claims and mechanistic findings.
>
> ## re: Scaling behaviors and Figure 3
>
> Thank you for your close inspection of our scaling trajectories. To clarify that the compute gap is non-constant for both AR and MDM, we provide an [updated Figure 3](https://imgur.com/a/FTNhP9m) which includes additional training runs to round out the frontiers.
>
> The empirical data demonstrates that the compute penalty of bytes is actually a dynamic efficiency ratio governed by the modeling objective:
> * At BPB=1.0, the AR compute gap is $\approx7.5\times$ (2.2e20 for Byte vs. 2.9e19 for BPE). However, at BPB=0.8 this gap shrinks to $\approx2.4\times$ (2.4e21 for Byte vs. 1.0e21).
> * The MDM byte models suffer a far greater penalty. At BPB=1.2 the compute gap is $\approx37.5\times$ (1.5e21 for Bytes vs. 4.0e19 for BPE). Scaling projections indicate that at BPB=1.0, the gap remains as high as $\approx20.8\times$ (1.0e22 vs. 4.8e20).
>
> These diverging trajectories are our core finding: with our refined scaling projections, we project AR parity at $F\approx1.4e22$, while MDM parity is projected two orders of magnitude later at $F\approx4.4e26$.
>
>
> ## re: disentangling data entropy from model performance.
>
> The reviewer correctly notes that data entropy impacts logprobs. To decouple inherent data entropy from modeling ability, we used the DEFLATE compression algorithm as a model-free proxy for structural information. Our studies reveal that while DEFLATE shows intra-block-8 data (preserving global order) is technically less compressible (higher entropy) than inter-block-8 sequences (preserving local contiguity), the AR models perform notably better on the intra-block-8 data. These results show that while maintaining local byte contiguity both preserves sequence compressibility and recovers model performance, maintaining larger-order global order gives an even stronger benefit: even with locally permuted bytes, models use causal data guarantees to recover model performance.
>
> ## re: Training curves (5.1) vs. scaling behavior (5.2)
>
> We clarify the analytical distinction between these two sections:
> * Section 5.1 (training progress) shows the learning curve of specific model configurations over time.
> * Section 5.2 (scaling laws) identify the isoFLOPs minima, i.e. the best performance for a given compute budget identified by optimizing the parameter-data split. This protocol, established by Hoffman et al. (https://arxiv.org/abs/2203.15556), allows us to compare the fundamental efficiency of different modeling paradigms regardless of individual run noise.
>
> ## re: Statistical Significance of Scaling Laws Points
>
> Thank you for raising this important point! We agree and have run several more training runs to round out the scaling laws plots. Please find our updated graphs here: [more rigorous isoFLOPs and scaling plots](https://imgur.com/a/FTNhP9m), which provide more support for the trends reported in our paper.
>
> ## re: “stable causal history” and BoolQ
>
> Thank you for the feedback about terminology. To clarify, “causal history” refers to the autoregressive factorization $p(x_t|x_{<t})$ where every predicted unit is conditioned on a stable, 100% unmasked prefix. While MDMs use positional embeddings, they lack a guaranteed causal context. Because a masked token must be resolved while its neighbors are simultaneously being denoised, the context remains unstable. This structural instability makes it harder for MDMs to resolve bytes. This may explain the stagnation in BoolQ (random chance is at 50%), as binary reasoning tasks are more sensitive to fine-grained semantic cues that MDM corrupts with the masking objective.
>
> ## re: Optimal BPB vs. Training Data plot
>
> We appreciate the proposal to observe scaling against data volume. While Figure 3 follows standard computational efficiency protocol, our Figure 2 (right) already serves as a sample efficiency analysis. Plotted against raw bytes of training data, AR byte and BPE models are practically overlaid, while MDM models retain a large efficiency gap, showing that Byte MDM suffers from relative sample inefficiency beyond mere computational overhead.
>
>
> Supporting media:
> * [more rigorous isoFLOPs and scaling plots](https://imgur.com/a/FTNhP9m)

---

> > ### Author Rebuttal · Reviewer_7a9z · 2026-04-04
> >
> > I thank the authors for their thoughtful response. While the updated Figure 3 has resolved some of my concerns, I still have follow-up questions about Section 7. Specifically:
> >
> > 1. I'm not convinced that DEFLATE is a good measurement of data entropy. One would expect that global random permutation (which, as the authors point out, destroy both local and global information) would have the highest entropy, and therefore the greatest magnitude of compressibility loss. However, the magnitude of compressibility loss for global random bytes (9.3%) is lower than that of inter-4 (14%), suggesting that global random is more compressible than inter-4, and therefore has less entropy.
> >
> > 2. The authors' claim in the rebuttal that "intra-block-8 data (preserving global order) is technically less compressible (higher entropy) than inter-block-8 sequences" doesn't seem to be true either. Intra-8 has compressibility loss of 6.4%, compared to Inter-8's 9%. If I understand this metric correctly, this would mean that Intra-8 is more compressible (lower entropy) than Inter-8, not less.
> >
> > 3. What sequence lengths did the authors use for Figure 5 and 6?
> >
> > I hope the authors can clarify these points.

---

> > > ### Author Response · Authors · 2026-04-05
> > >
> > > We are glad that our previous response helped resolve your concerns regarding the scaling trajectories. We are grateful for your close read of section 7; your observation helped us catch a clerical error in our original manuscript that we are eager to correct.
> > >
> > >
> > > ## re: DEFLATE and Permutation compressibility typo
> > > You are correct that the compressibility loss values for the Global Byte and Intra-8 permutations were reported incorrectly in our original draft due to a dropped tens digit during final composition. We apologize for this oversight. The corrected values are as follows:
> > >
> > >
> > >  | $\pi$ Strategy | Visualization | Compr. Loss (Corrected) | Compr. Loss (Original Draft) |
> > >  | :--- | :--- | :---: | :---: |
> > >  | **Global Byte** | `_mus isde ofno dlif` | **19.3%** | 9.3% |
> > >  | **Intra-8** | `ifdfsiuo noeld_ms` | **16.4%** | 6.4% |
> > > | **Inter-4** | `n_mo diff dels usio` | 14.0% | 14.0% |
> > >  | **Inter-8** | `n_models diffusio` | 9.0% | 9.0% |
> > >
> > >
> > > With these corrected numbers, the data matches the intuition you noted: global byte and  intra-8 (which destroy local n-gram contiguity) represent the greatest increase in data entropy.
> > > These corrected numbers actually provide even stronger support for our central thesis. We find it particularly interesting that even though intra-8 sequences are significantly more difficult for a model-free compressor to resolve (16% loss)  than inter-8 data (9% loss), the AR model manages to recover significantly more performance on that harder data, suggesting that the model isn’t just mirroring the statistical entropy of the data, but is leveraging the stable causal history as a powerful inductive bias to resolve data predictability.
> > >
> > >
> > > ## Multi-compressor validation
> > > We take your point that DEFLATE may be an imperfect proxy for data entropy given its reliance on a specific sliding-window inductive bias. To address this, we have expanded our analysis to include Zstd, LZMA (high-compression algorithm optimized for long-range repetitions), and PPMd Order-8 (a Markovian predictor closer to a statistical language model and particularly good at compressing text data).
> > >
> > >  | $\pi$ Strategy | DEFLATE (v6) | LZMA (v6) | Zstd (v3) | PPMd (O8) |
> > >  | :--- | :---: | :---: | :--- | :---: |
> > >  | **Global BPE** | 6.9% | 6.0% | 6.6% | 4.8% |
> > >  | **Global Byte** | 19.3% | 18% | 14.9% | 24.2% |
> > >  | **Intra-8** | 16.4% | 14.9% | 13.1% | 20.3% |
> > >  | **Inter-4** | 14% | 14.4% | 13.3% | 15.9% |
> > >  | **Inter-8** | 9% |  9% | 9.5% | 8.7% | 10% |
> > >
> > >
> > > Across all four compression engines, the trend remains robust: the destruction of local contiguity in Intra-8 and global shuffles produces the most significant increase in data entropy.
> > >
> > >
> > > ## re: Sequence length for permutation experiments.
> > >
> > > For the permutation experiments in Figures 5 and 6, we maintained the same information-equivalent 4x context length ratio used throughout our studies to ensure fair data content across representations. Specifically, we used $L_{BPE}=1024$ and $L_{Byte}=4096$.
> > >
> > >
> > > We hope these follow-ups address your concerns and reinforce the value of our empirical and mechanistic findings. We will ensure the final manuscript reflects these corrected values and the resulting clarified analysis. Thank you again for your rigorous feedback!

---

### Decision · Program_Chairs · 2026-04-30

**Decision:**

Accept (regular)

**Comment:**

This paper studies the interaction between tokenization granularity and modeling objective by comparing autoregressive and masked-diffusion language models under both byte-level and BPE representations. The main finding is an objective-dependent efficiency gap: byte-level AR models partially recover subword structure at scale, whereas byte-level masked diffusion remains substantially less efficient. Reviewers generally found the question timely, the methodology careful, and the compute-matched setup to be a major strength.

The main discussion centered on interpretation and generality. One reviewer questioned whether the AR-byte results justify stronger "near parity" language, since a nontrivial compute gap may persist at matched performance. Other concerns were that the sequence-length normalization may introduce a confound and that the negative conclusion for masked diffusion may be too broad given the limited exploration of alternative MDM designs. These are real caveats, but the rebuttal appears to have resolved most of them to leave the overall assessment positive.

Overall, I recommend weak accept. The final version should sharpen the wording around the AR-byte result and more clearly separate firmly supported empirical findings from the more interpretive mechanistic claims. I also encourage the authors to tight the claims a bit since the inefficiency of MDM at byte-level modeling might be an issue with the existing masking strategies -- a further improvement on it (eg consecutive masking) might help.